# Relative Risk Functions for Estimating Excess Mortality Attributable to Outdoor PM$_{2.5}$ Air Pollution: Evolution and State-of-the-Art

**Richard Burnett [1,\*] and Aaron Cohen [1,2]**

[1]   Institute for Health Metrics and Evaluation, University of Washington, Seattle, WA 98121, USA; acohen@healtheffects.org

[2]   Health Effects Institute, Boston, MA 02110-1817, USA

\*   Correspondence: rtburnett1@gmail.com

**Abstract:** The recent proliferation of cohort studies of long-term exposure to outdoor fine particulate air pollution and mortality has led to a significant increase in knowledge about this important global health risk factor. As scientific knowledge has grown, mortality relative risk estimators for fine particulate matter have evolved from simple risk models based on a single study to complex, computationally intensive, integration of multiple independent particulate sources based on nearly one hundred studies. Since its introduction nearly 10 years ago, the integrated exposure-response (IER) model has become the state-of-the art model for such estimates, now used by the Global Burden of Disease Study (GBD), the World Health Organization, the World Bank, the United States Environmental Protection Agency's benefits assessment software, and scientists worldwide to estimate the burden of disease and examine strategies to improve air quality at global, national, and sub-national scales for outdoor fine particulate air pollution, secondhand smoke, and household pollution from heating and cooking. With each yearly update of the GBD, estimates of the IER continue to evolve, changing with the incorporation of new data and fitting methods. As the number of outdoor fine particulate air pollution cohort studies has grown, including recent estimates of high levels of fine particulate pollution in China, new estimators based solely on outdoor fine particulate air pollution evidence have been proposed which require fewer assumptions than the IER and yield larger relative risk estimates. This paper will discuss the scientific and technical issues analysts should consider regarding the use of these methods to estimate the burden of disease attributable to outdoor fine particulate pollution in their own settings.

**Keywords:** global burden of disease; integrated exposure-response; global mortality exposure model

---

## 1. Introduction

Exposure to outdoor fine particulate matter (PM$_{2.5}$) was the eighth leading risk factor for global mortality in 2017, contributing to 2.9 (95% confidence interval: 2.5, 3.4) million deaths, largely from non-communicable cardiovascular and respiratory diseases [1]. The preponderance of global excess deaths attributable to past exposure to outdoor PM$_{2.5}$ has occurred in low- and middle-income countries (LMICs) in East Asia, South Asia, Africa, and the Middle East, with 52% occurring in China and India alone. In these regions, levels of outdoor PM$_{2.5}$ increased over the past several decades while, over the same period, levels markedly declined in high-income countries of Europe and North America, reaching current levels that are as much as an order of magnitude lower than in some of the most polluted LMICs [1–3] (Figure 1).

Average Annual Population−Weighted PM$_{2.5}$ Concentrations in 2017

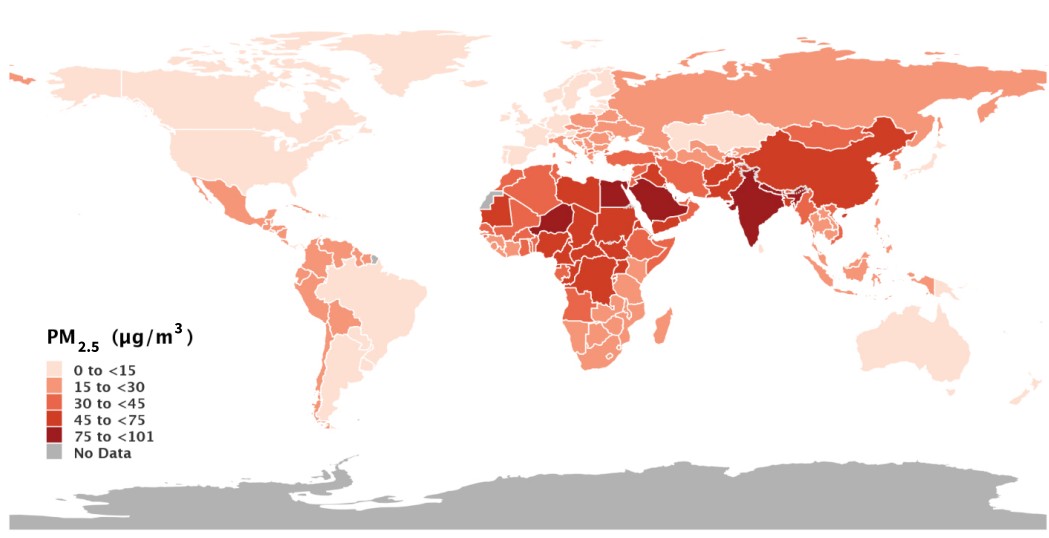

State of Global Air

**Figure 1.** Average annual population-weighted fine particulate matter (PM$_{2.5}$) concentrations in 2017. Source: Health Effects Institute. 2019. State of Global Air 2019. Accessed on 9 May 2020. Data source: Global Burden of Disease Study 2017. IHME, 2018.

In recognition of this major public health threat, the United Nations (UN) Sustainable Development Goals now call for substantial reductions in mortality caused by exposure to outdoor PM$_{2.5}$ by 2030 [4]. Actions are being taken in some major LMICs, such as China, where an aggressive plan started in 2013 appears to have had some initial success in reducing PM$_{2.5}$ levels, and India where the government is beginning to implement a more aggressive air quality management program [4]. Furthermore, in high-income countries in Europe and North America there are continuing efforts to reduce already low levels further, given considerable evidence that increased risk of mortality persists even at low levels of outdoor exposure to PM$_{2.5}$ [5,6]. As a result, there is a critical and continuing need to both quantify current excess mortality attributable to past exposure to outdoor PM$_{2.5}$ and estimate the benefits that could accrue from reducing exposure at global, national, and local levels.

Exposure to outdoor PM$_{2.5}$ in both the short- and long-term cause increased morbidity and mortality from cardiovascular and respiratory disease, lung cancer, and lower respiratory infections; however, long-term exposure, on the order of years, has by far the largest impacts on public health [7–9]. Long-term exposure to outdoor PM$_{2.5}$ reduces life expectancy via its effects on disease occurrence and mortality [10]. For this reason, the burden of disease attributable to outdoor PM$_{2.5}$ has been estimated using the results of epidemiologic cohort studies which observe large populations over years and relate the long-term outdoor exposure of cohort members to their observed mortality from all-natural (non-accidental) and/or specific causes of death.

The epidemiologic evidence has been used to compute various metrics to quantify the burden of disease attributable to outdoor PM$_{2.5}$, but the most commonly reported is the number of excess deaths attributable to exposure. It can be interpreted as the number of deaths in a given year attributable to past exposure. Although analysts often refer to such estimates as premature deaths attributable to exposure, it is important to note that quantifying prematurity requires the use of a time-based metric such as disability-adjusted life-years (DALYs) or years of life lost (YLL). These issues are discussed in greater detail elsewhere [11,12].

It is also important to distinguish between attributable deaths, deaths that occurred due to past exposure, and avoidable deaths, deaths that would not be expected to occur in the future should

exposure be reduced [12]. In particular, analysts wishing to quantify the future reduction in mortality due to an action taken to reduce exposure need to account for future changes in underlying mortality rates and population size and age structure because these factors may exert a larger effect on mortality than the predicted changes in outdoor $PM_{2.5}$ exposure [13,14]. This same caveat applies to analyses of time trends in attributable mortality due to past exposure, whose interpretation should account for past trends in underlying mortality rates and demographics [15].

Estimating excess deaths attributable to outdoor $PM_{2.5}$ air pollution requires four major inputs: estimates of the distribution of population-weighted exposure; specification of a counterfactual, cleaner, level of exposure below which no increased risk of mortality is assumed to exist; estimators of the relative risk, termed exposure-response (E-R) functions, across the entire exposure distribution from the highest level to a cleaner, counterfactual level; and estimates of baseline mortality. Estimates of exposure and relative risk are then combined to estimate a population attributable fraction (PAF), the proportion of deaths attributable to exposure above the counterfactual level. The baseline deaths are then multiplied by the PAF to estimate the excess deaths attributable to exposure [15].

The integrated exposure-response (IER) model, developed for use in the Global Burden of Disease Study (GBD), has become the state-of-the art exposure-response model for estimating the $PM_{2.5}$ mortality relative risk since its introduction nearly a decade ago [16]. By integrating mortality relative risk estimates from other $PM_{2.5}$ combustion sources, including secondhand smoking, household burning of solid fuels, and active smoking, the IER made it possible to estimate $PM_{2.5}$ relative risks across the entire global range of exposure, including highly polluted areas in East and South Asia where epidemiologic studies are lacking [17]. The IER is now used by the GBD, the World Health Organization's (WHO) benefits assessment software (AirQ+), the World Bank, the United States Environmental Protection Agency's (US EPA) benefits assessment software (BENMAP), and scientists worldwide to estimate excess mortality and examine strategies to improve air quality at global, national, and sub-national scales. With each yearly update of the GBD, estimates of the IER continue to evolve, changing with the incorporation of new data and fitting methods. As the number of outdoor fine particulate air pollution cohort studies has grown, including recent estimates of high levels of $PM_{2.5}$ pollution in China, new relative risk estimators based solely on outdoor fine particulate air pollution evidence have been proposed which require fewer assumptions than the IER and yield larger outdoor fine particulate relative risk estimates [18].

Analysts seeking to quantify excess mortality from outdoor $PM_{2.5}$ now have a more diverse menu of estimators from which to choose, and, as a result, the scientific, public health, and policy communities have been presented with varying estimates of excess mortality. This paper discusses the scientific and technical issues that underlie the currently available relative risk estimators and the issues that analysts should consider regarding the use of these methods. Their relative utility depends on the analytic objectives which may include not only estimation of attributable mortality in a given place or time, but also comparison with attributable mortality from other risk factors, evaluation of actions undertaken to improve air quality, and quantification of economic benefits of $PM_{2.5}$ reduction [2,17–19]. Choice of relative risk estimator may also affect the public health interpretation and application of the estimates of excess mortality they provide [20].

## 2. Estimation of Attributable Deaths Due to $PM_{2.5}$ Exposure

In this paper we focus on approaches to determine the number attributable deaths, $\Delta AD$, attributable to the difference in actual outdoor $PM_{2.5}$ exposure, and cleaner, counterfactual conditions such as that which might have occurred due to actions taken to improve air quality, $cf$, over a fixed time period. To do this, one preforms the calculation

$$\Delta AD = pop \times M \times PAF_{\beta}(PM_{2.5}, cf)$$

where *pop* denotes the size of the target population, $M$ is the mortality rate for that target population, and $PAF_\beta(PM_{2.5}, cf)$ is the population attributable fraction associated with the change in pollution concentrations, and is defined by

$$PAF_\beta(z_C, z_F) = 1 - \frac{1}{R_\beta(PM_{2.5}, cf)}$$

where $R_\beta(z_C, cf)$ denotes the relative risk function indexed by a parameter vector $\beta = (\beta_1, \ldots, \beta_P)$ that characterise its shape and magnitude. It represents the ratio of the probability of a death if a person is exposed to concentration $z_C$ divided by the probability of a death if they were exposed to concentration $cf$. In order to perform this calculation, one needs to specify the mathematical form of the relative risk function and determine the values of the parameter vector.

The first mathematical form of the relative risk function is the log-linear model:

$$LL_\beta(z_C, z_F) = \exp\{\beta \times (PM_{2.5} - cf)\}.$$

It is called the log-linear model since the logarithm of the relative risk is linear to changes in concentration. Note, this mathematical form suggests that the relative risk is a function of only the difference in concentrations examined, $PM_{2.5} - cf$, not their individual magnitudes.

The US EPA has used this model to estimate benefits due to reductions in air pollution in the United States [21]. In this case, the US EPA relied on the value of $\beta$ from a single cohort study of long-term exposure to $PM_{2.5}$ and mortality conducted by the American Cancer Society [22]. However, the use of the log-linear model poses a particular problem for assessments in any locale with high levels of outdoor $PM_{2.5}$. In particular, extrapolating log-linear model coefficients derived from studies in low-exposure, high-income, Western countries to much greater levels of outdoor $PM_{2.5}$ results in implausibly large estimates of relative risk and attributable deaths in LMICs.

GBD 2004 was the first time the GBD included outdoor $PM_{2.5}$ as a risk factor [23]. They used a log-linear model relative risk estimate from the American Cancer Society cohort for cardiopulmonary deaths [22] to estimate attributable deaths in urban areas worldwide. In order to limit the magnitude of the relative risk at high concentrations, they assumed no additional risk beyond 30 μg/m$^3$. This highly conservative assumption highlighted the need for evidence about the magnitude of the relative risk and the shape of its exposure-response function at these higher levels.

## 3. Estimators of the $PM_{2.5}$ Mortality Relative Risk over the Global Concentration Range

### 3.1. The Integrated Exposure-Response (IER) Relative Risk Model

Although the number of outdoor fine particulate air pollution cohort studies increased since 2004, all of the new studies have been conducted at low levels of exposure in high-income countries; thus, the GBD still faced the same problem as it planned an updated set of estimates for GBD 2010 [15,18]. In those studies, the estimated annual-average population-weighted exposure rarely exceeded 50 μg/m$^3$, with most below 30 μg/m$^3$.

A solution was suggested from an analysis by Pope and colleagues [24,25] which linked relative risks of cardiovascular and lung cancer deaths to other sources of $PM_{2.5}$, including those from secondhand and active smoking, which result in much higher $PM_{2.5}$ exposure than observed outdoors. They placed outdoor $PM_{2.5}$ air pollution as well as secondhand and active smoking relative risks on the same dose scale: total inhaled $PM_{2.5}$ mass. Graphically displaying all the relative risks on the total mass scale revealed a non-linear shape with the change in relative risk decreasing as total mass increased. However, this non-linear shape was much more pronounced for cardiovascular mortality than for lung cancer [25].

The analysis presented by Pope and colleagues [24,25] paved the way for the GBD to develop the integrated exposure-response (IER) relative risk function. In addition to the three types of $PM_{2.5}$

exposures considered by Pope and colleagues, the GBD added a fourth type: household pollution from heating and cooking sources [17]. This enabled the GBD to estimate attributable mortality from three combustion-derived risk factors, outdoor fine particulate air pollution, secondhand smoking, and household burning of solid fuels, using a single risk function. Exposures from all four sources were calculated on an equivalent scale of 24-h average concentration, the scale used for outdoor, household, and secondhand smoking studies. Only active smoking exposure was converted from total mass of inhaled particles to an equivalent daily outdoor concentration. Active smoking concentrations were assigned 667 $\mu g/m^3$ per cigarette smoked, and with cohort studies often reporting 20 or more cigarettes/day smoked, the IER covered any observed concentration in the outdoor range, including those in the much higher household range of thousands of $\mu g/m^3$ [17].

Integrating these four types of $PM_{2.5}$ exposures required several important assumptions [17]. Specifically, the relative risk at any concentration was independent of the type of $PM_{2.5}$, and only dependent on the magnitude of the total exposure from all types together. The relative risk was also independent of the dosing rate within a day. For example, active cigarette smoking consists of a sequence of inhalations for each cigarette, potentially repeated several times a day when multiple cigarettes are smoked. Large doses of particles are inhaled during cooking periods a few times per day for household pollution, and for times when a non-smoker is in the home of a smoker for secondhand smoke. Exposure to outdoor-based $PM_{2.5}$ is likely to be more evenly distributed throughout the day.

The IER used for GBD 2010 had the mathematical form [17]:

$$IER(z) = 1 + \vartheta\left(1 - \exp\left(-\omega z^{\delta}\right)\right)$$

where $z = \max(0, \ PM_{2.5} - cf)$. The IER can take on a variety of shapes, including supra, sub, and near linear. The unknown parameters $(\vartheta, \omega, \delta)$ are estimated using non-linear regression methods [17].

The GBD includes a counterfactual uncertainty distribution as part of the parameter estimation process. This distribution represents current uncertainty about the lowest levels of exposure that cause increased mortality. GBD 2015 defined the counterfactual uncertainty distribution as a uniform random variable with a lower bound of 2.4 $\mu g/m^3$ and an upper bound of 5.9 $\mu g/m^3$ [15]. These values were based on the average minimum/5th percentile of cohort studies whose 5th percentile was at or below that of the American Cancer Society cohort [22], which was 8.2 $\mu g/m^3$ [26]. Since the specification of the counterfactual is an integral part of the IER estimation it cannot be changed by potential users.

The GBD also adjusts the IER by age for the two cardiovascular causes of death, ischemic heart disease (IHD) and stroke, with the logarithm of the relative risk declining in a linear manner with age [15]. For GBD 2010, five-year age groups, starting at age 25, were used up to the 80–84 age group. An older age group of 85+ years was also included. For GBD 2016, and subsequent GBD versions, age groups for 85–89, 90–94, and 95+ years were added.

The GBD updates burden estimates on a yearly basis. For outdoor fine particle air pollution, all elements of the burden analysis are updated, including baseline mortality rates, global $PM_{2.5}$ estimates, and the IER. Updates to the IER include both the addition of new studies and statistical methods to estimate the IER parameters. A comprehensive assessment of the influence of these updates is provided in the Methods Appendices of the GBD capstone papers [1].

*3.2. Relative Risk Models Using Only Ambient Air Pollution Cohort Studies*

Continuing uncertainty regarding the assumptions that underlie the IER, discrepancies between its predictions and some cohort study results, and practical difficulties faced by analysts outside the GBD in applying the IER in their own settings, have led to a search for alternative risk models [18]. At the same time, cohort studies examining the association between outdoor $PM_{2.5}$ concentrations and mortality, especially in Canada, the United States, and Western Europe, and for the first time Chinese studies [27,28], have proliferated, making such new models more feasible. To date, however, there has

not been a consistent method to examine the shape of this association within each cohort, nor methods to pool these shapes among cohorts, providing a common relative risk model for burden assessment.

### 3.2.1. Shape Constrained Health Impact Function (SCHIF)

Nasari et al. [29] proposed a method to model the $PM_{2.5}$–mortality association using data within any specific cohort. Their model, the Shape Constrained Health Impact Function (SCHIF) has the form

$$logSCHIF(z) = \theta f(z)l(z)$$

where $l(z) = (1 + \exp(-(z - \mu)/\tau r))^{-1}$ is a logistic function in concentration. Here, $(\theta, \mu, \tau)$ are unknown parameters to be estimated from the data, with $r$ as the range in the translated exposure: $z = PM_{2.5} - \min(PM_{2.5})$, such that $logSCHIF(0) = 0$. The function $f(z)$ can take two forms: $f(z) = z$ (linear) and $f(z) = \log(z + 1)$ (log). The SCHIF is similar to the log-linear model (LL): $logLL(z) = \beta z$, in the sense it is a function of concentration times of a parameter by writing: $logSCHIF(z) = \theta T(z)$, where $T(z) = f(z)l(z)$ is a specific transformation of concentration. The linear form $f(z) = z$ can model both linear and sub-linear associations, while the log form $f(z) = \log(z + 1)$ can model supra-linear associations with mortality. Both forms can accommodate "S" shaped functions through $l(z)$.

Burnett and colleagues [18] fit the SCHIF to 15 of the world's largest cohorts including a cohort in China where concentrations exceeded 84 $\mu g/m^3$ [28]. The focus of this work was on all non-accidental causes of death as they have increased statistical power to distinguish one shape from another. The much lower death counts for the specific causes of death examined by GBD were deemed to have limited statistical power for this purpose. A variety of shapes were observed, however, 12 of the 15 cohorts displayed a supra-linear form.

### 3.2.2. Global Exposure Mortality Model (GEMM)

A common relative risk model among the 15 cohorts with cohort-specific SCHIF models and an additional 26 cohorts where it was assumed, like the GBD, that the association was linear, for all causes of death and the five specific causes examined by GBD 2015 was proposed [18]. The common model, known as the Global Exposure Mortality Model (GEMM) has the form

$$logGEMM(z) = \theta \log(z/\alpha + 1)/(1 + \exp(-(z - \mu)/\tau r))$$

Here, an additional parameter $\alpha$ was added to make the SCHIF more flexible. Parameters in the GEMM were constrained such that the change in relative risk for higher concentrations declined as concentration increased, thus limiting the magnitude of the relative risk for the most polluted parts of the globe where no studies have been conducted. The attributable number of deaths due to $PM_{2.5}$ exposure globally was about twice that predicted by the IER, in part because the GEMM models all-natural cause mortality and in part because the IER incorporates additional types of exposure, such as active smoking, that have lower relative risks per unit $PM_{2.5}$ than ambient air pollution [17,24,25].

## 4. Recommendations for Conducting Burden and Benefits Assessments

### *4.1. Comparison of the Relative Risk Modeling Approaches*

To date, there are three types of relative risk models proposed for assessing the population mortality burden due to outdoor $PM_{2.5}$ exposure: LL, IER, and GEMM. Each of these model specifications has strengths and limitations which have differing implications depending on the specific analytic objectives.

### 4.1.1. Comparison of Relative Risk Estimates and Population Attributable Fractions

We illustrate the differences in these models by first comparing the relative risks for each model at lower outdoor concentrations (Figure 2) and over the global range (Figure 3). The LL relative risk for $PM_{2.5}$ concentrations below 30 $\mu g/m^3$ is displayed in the upper left-hand panel (Figure 2)

for all ages above 25 years. Here, the counterfactual is set to 2.4 μg/m³ in order to make a direct comparison with the GEMM counterfactual. We denote these models as: non-communicable disease (NCD) + lower respiratory infection (LRI), since the GEMM has been applied to all non-communicable diseases plus lower respiratory infection mortality rates as this set of diseases represents over 99% of all non-accidental mortality in the cohorts [18]. This avoids applying the GEMM in locations with substantial communicable disease mortality.

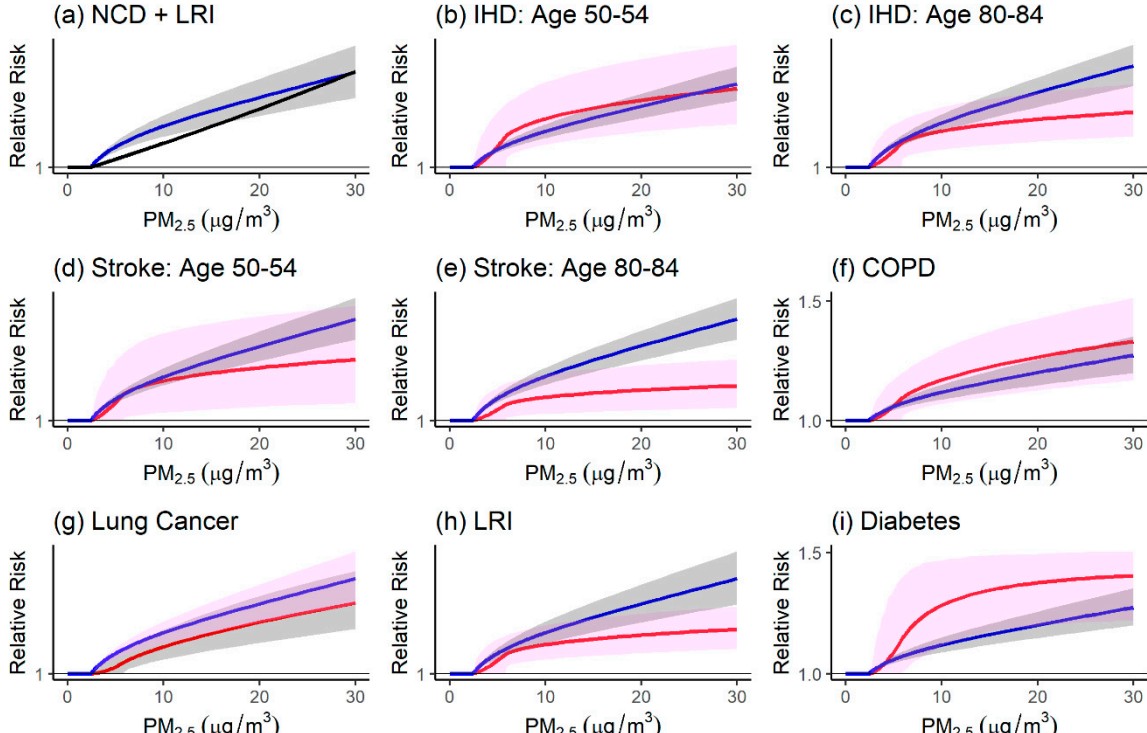

**Figure 2.** Relative risk by PM$_{2.5}$ concentrations below 30 μg/m³: (**a**) NCD + LRI GEMM (solid blue line) and LL model (solid black line). GEMM 95% confidence interval presented as grey shaded area. IER relative risk (solid red line) presented by cause of death: (**b**) IHD Age 50–54; (**c**) IHD Age 80–84; (**d**) Stroke Age 50–54; (**e**) Stroke Age 80–84; (**f**) COPD; (**g**) Lung Cancer; (**h**) LRI; (**i**) Diabetes. IER 95% confidence intervals presented as pink shaped areas. NCD = non-communicable disease; LRI = lower respiratory disease; GEMM = Global Exposure Mortality Model; LL = log-linear; IER = integrated exposure-response.

The GEMM is supra-linear with a larger increase in the relative risk at lower concentrations compared to the LL model. The GEMM is also compared to the IER models for the six specific causes of death examined by GBD2017 [15]. Both IHD and stroke are adjusted by age with the relative risk declining with increasing age. To illustrate this age-dependency we present results for the 50–54 years old and 80–84 years old age groups. Note the GEMM is also age-adjusted. The NCD + LRI GEMM relative risk tends to be lower for chronic obstructive pulmonary disease (COPD), LRI, and diabetes and larger for IHD age 80–84, stroke, and lung cancer.

A similar comparison is presented in Figure 3 for the entire global range that has been estimated to be as high as 300 μg/m³ [30]. The LL model relative risks start to diverge from the GEMM relative risks at approximately 50 μg/m³ (upper left-hand panel, Figure 3). The GEMM shows the relationship between relative risk and concentration over the exposure range of the cohort studies but is constructed in such a manner as to model the change in relative risk on the order of a logarithmic function above the observed exposure range. Thus, the GEMM relative risks are sustainably lower than the LL relative risks above 50 μg/m³. Note that the LL model estimates a very large relative risk of 13.7 at 300 μg/m³. This biologically implausible value for NCD + LRI deaths motivated the formulation of the GEMM to

limit relative risk predictions beyond the observed exposure range of the cohorts. However, the GEMM relative risks are much higher than the IER relative risks over the higher concentration range. This is due to the inclusion of information in the IER on secondhand smoke, household pollution, and active smoking, whose relative risks per unit change in $PM_{2.5}$ are much lower than those of ambient air pollution [17].

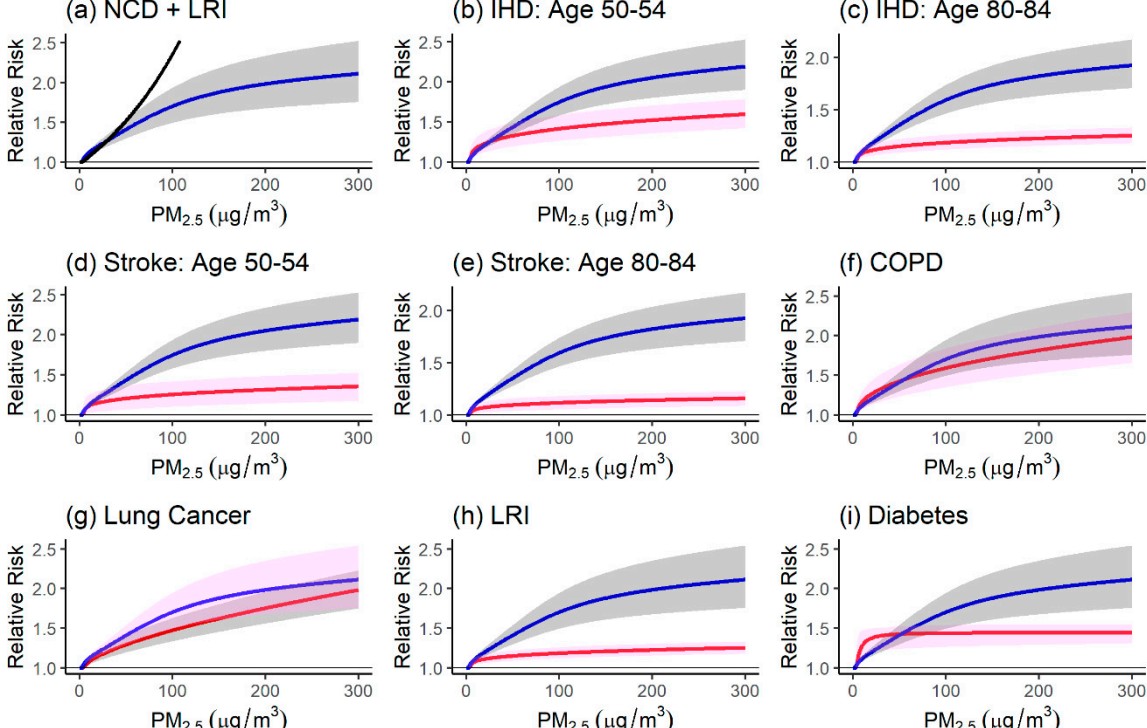

**Figure 3.** Relative risk by $PM_{2.5}$ concentration over the global concentration range: (**a**) NCD + LRI GEMM (solid blue line) and LL model (solid black line). GEMM 95% confidence interval presented as grey shaded area. IER relative risk (solid red line) presented by cause of death: (**b**) IHD Age 50–54; (**c**) IHD Age 80–84; (**d**) Stroke Age 50–54; (**e**) Stroke Age 80–84; (**f**) COPD; (**g**) Lung Cancer; (**h**) LRI; (**i**) Diabetes. IER 95% confidence intervals presented as pink shaped areas. NCD = non-communicable disease; LRI = lower respiratory disease; GEMM = Global Exposure Mortality Model; LL = log-linear; IER = integrated exposure-response.

We next compared the PAFs estimated from the three relative risk models. We calculated the ratio of each of the six GBD causes of death mortality rates to those of NCD + LRI globally for the two age groups for the year 2017. We then constructed a weighted average of the six causes of death PAFs with weights defined by these ratios. This provides a direct comparison of the total mortality impact of the six causes of death together compared to the NCD + LRI GEMM (Figure 4). The sum of mortality rates of the six causes of death for the 50–54 years old age group was 48% of NCD + LRI deaths and 60% for the 80–84 years old age group. We also calculated the PAF for the LL model for all adult ages for comparison purposes.

The PAF for the 50–54 age group (solid lines) was greater than for the 80–84 age group (dashed line) for both the GEMM and IER since relative risk was modeled to decline with increasing age in both model formulations. The GEMM PAF (blue lines) is larger than either the LL PAF (solid black line) and the IER PAF (red lines) for lower concentrations (left-hand panel, Figure 4). For higher concentrations (right-hand panel), however, the LL PAF is much larger, reaching almost unity at $300\ \mu g/m^3$, suggesting all mortality in a population with this extreme exposure is caused by $PM_{2.5}$. In contrast, the IER PAF at $300\ \mu g/m^3$ is approximately 10%. The GEMM PAF is intermediate, at 45%, between the IER and LL PAFs.

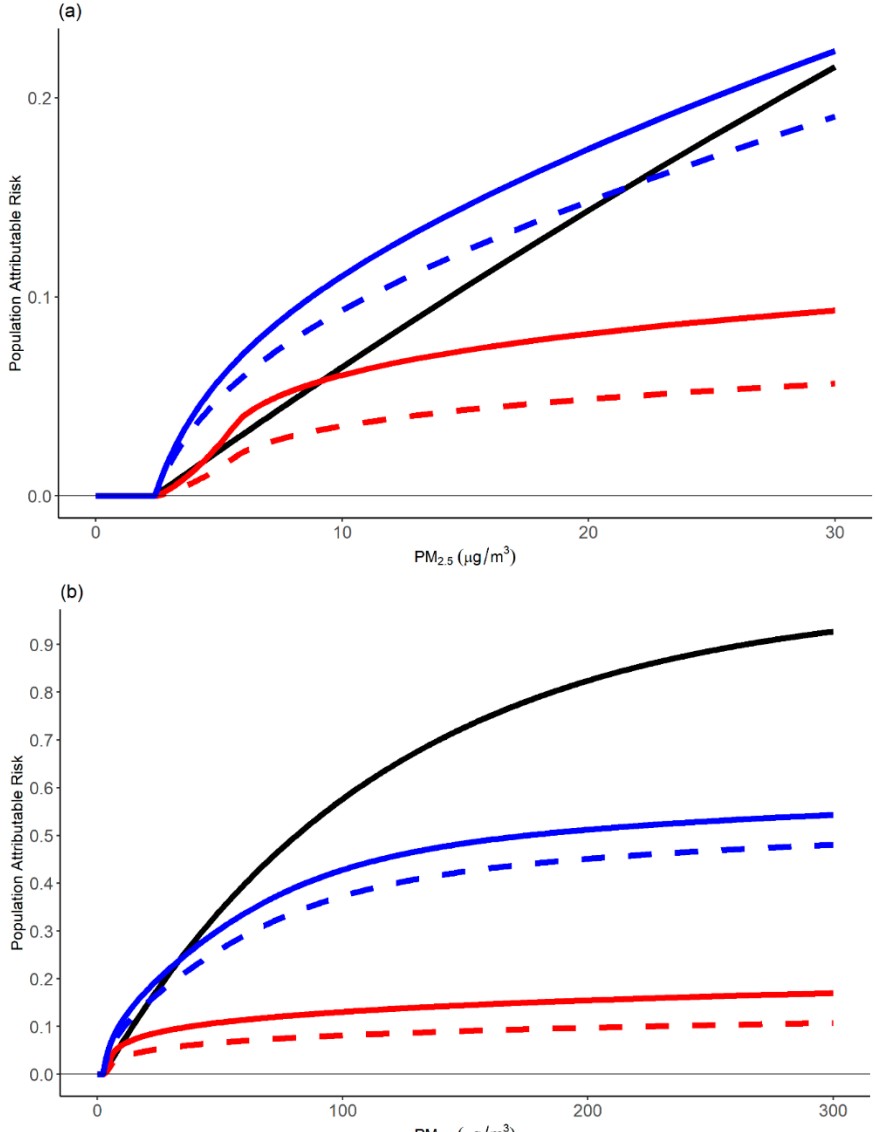

**Figure 4.** Population attributable fraction (PAF) of NCD + LRI GEMM (blue lines), IER (red lines), and LL (black line) over low concentration range (**a**) and global range (**b**). NCD + LRI GEMM and IER PAFs are presented for 50–54 years old (solid lines) and 80–84 years old (dashed lines) age groups.

Comparison of the PAFs clearly demonstrate why global attributable mortality based on the NCD + LRI GEMM was much larger than that based on the summation of the attributable deaths from specific causes using the IER [18]. This difference is based on the combination of differences in relative risk estimates and the fact that the six GBD causes of death only represent half that due to NCD + LRI. Since the NCD + LRI mortality rates are much larger than the sum of the six specific causes of death mortality rates, so are the estimates of attributable deaths.

4.1.2. Strengths and Limitations of Modeling Approaches

Strengths of the LL model are that it only uses information on risk and exposure from cohort studies of ambient air pollution. Its limitations are that the linear association between exposure and mortality may not hold for all outcomes. In addition, LL-based estimates of the relative risk at the highest global concentrations approach that of the relative risk of active smoking and lung cancer. These biologically implausible estimates have limited the LL model use in global burden assessments, but many analysts are primarily interested in estimating attributable mortality from outdoor $PM_{2.5}$ in

national, regional or local settings with much lower concentrations, such as in Canada, the United States, and Western Europe, locations where most cohort studies of outdoor fine particulate air pollution and mortality have been conducted. In these locations, the LL model could be an option, in particular in conjunction with more flexible model forms, such as the IER or GEMM.

The IER model extends the LL model to cover the entire global concentration range by adding information on risk and mortality from cohort studies of other types of fine particle exposures such as secondhand smoke, household pollution, and active smoking. These additional exposures are assigned much higher $PM_{2.5}$ concentrations in the model fitting process than those currently observed in outdoor fine particulate air pollution studies. This modeling approach does not therefore require extrapolation of exposure from relatively low levels to much higher concentrations observed globally. However, in order to integrate the relative risks for the different particle sources, estimates of exposure from the different sources are required. In addition, the assumption that the relative risk of each of these sources is identical at any concentration is also required. However, these assumptions have not been formally tested against empirical evidence.

The GEMM was developed to extend the LL model to potentially non-linear associations between exposure and mortality observed both within cohorts and pooling these associations among cohorts. As with the LL model, the exposure ranges of the cohort studies were limited, generally below 30 $\mu g/m^3$. One cohort of Chinese men [28] was available with exposures up to 84 $\mu g/m^3$, thus greatly extending the observed concentration range. However, the GEMM was highly sensitive to inclusion of this cohort, highlighting the need for additional cohorts at higher levels. In order to limit the magnitude of the GEMM at high global concentrations, restrictions of the shape were enforced, resulting in a supra-linear shape with the highest relative change at the lowest concentrations. We thus recommend caution in interpreting GEMM predictions at these high levels, in particular if they exceed those of other known risk factors such as active smoking.

*4.2. Should Estimates Be Based on All-Natural Cause or Cause-Specific Mortality?*

Analysts have estimated the mortality attributable to ambient $PM_{2.5}$ in terms of both all-natural cause and cause-specific mortality. All-natural cause mortality rates can be thought of as a weighted sum of the underlying non-accidental cause-specific rates, and the relative contribution of specific causes may differ in space and time. For example, some low-income countries have higher rates of mortality from infectious diseases and the proportional contribution of mortality from stroke and heart disease to total mortality has risen in LMICs over time [31]. Since the GBD's objective is to estimate the burden of disease in all countries, over time it has based its estimates of risk factor-attributable burdens, including the IER, on specific causes of death where there is strong evidence for an effect of exposure to outdoor $PM_{2.5}$ [2].

The GBD currently uses six specific causes of death to estimate attributable mortality: ischemic heart disease, stroke, chronic obstructive pulmonary disease, lung cancer, lower respiratory infections, and type II diabetes based on an assessment of the strength of the epidemiologic evidence [1]. Separate IER models are created for each cause of death. New causes are added based on yearly assessments of the emerging evidence using systematic review techniques. Air pollution epidemiologists, however, have frequently estimated effects on all-natural cause mortality in epidemiologic studies, due in part to concerns with the availability and quality of cause-specific mortality ascertainment in many locales. The GEMM estimates the relative risk of non-accidental causes of death because this outcome has been used by analysts and regulatory agencies for risk and burden assessment. GEMMs were also created for five of the six causes of death currently used in the GBD.

Globally, the six causes of death included in the current GBD estimates represent 41% of non-accidental mortality and 52% of non-communicable disease plus lower respiratory infection mortality [31]. This raises concerns that using only these six causes could underestimate total mortality burden if outdoor fine particulate air pollution increases mortality from other causes of death. In fact, in a comparative analysis between burden estimated by the IER using only the five causes, and the

GEMM using all non-accidental deaths, the IER-based burden estimate was half that of the GEMM. However, extending the GEMM based on studies in Western countries with a specific distribution of causes of death, globally, has limitations, since other countries' cause of death distributions can be markedly different [32]. Burden assessments that use specific causes of death are not subject to this limitation.

## 5. Recommendations and Future Directions

Differences in estimates of attributable mortality produced using current models reflect our uncertainty about the true size of the $PM_{2.5}$ mortality relative risks at high levels of exposure [27,32]. The IER and GEMM have been evaluated with respect to their efficacy in conducting global assessments and we concur with these assessments that the IER remains a useful, indeed the preferred, model for global assessments [27,30]. Therefore, we suggest that global assessments be conducted using the IER, with the disease specific and non-accidental GEMMs used in sensitivity analyses to model uncertainty in estimates of attributable deaths. GEMMs using only non-communicable diseases and lower respiratory infection might also be used as these disease groups represented over 99% of deaths in the cohort studies.

Mortality relative risk models for $PM_{2.5}$ are continuing to evolve. Currently, the GBD is considering changes to the IER, including removal of active smoking, new methods of exposure estimation, using more flexible spline models in their fitting procedures, and allowing users to define their own counterfactual uncertainty distribution. Future updates of the GBD estimates will also consider growing evidence regarding exposure to $PM_{2.5}$ and other important causes of mortality including adverse reproductive outcomes and neurologic diseases and add these causes if the evidence warrants it [8]. These changes are expected to significantly narrow the gap between burden estimates based on the IER and GEMM. Changing the form of the GEMM to specially include the LL model at lower exposures would allow the GEMM to conform to the widely used LL model estimates in Western countries.

However, the most critical need is for new studies to reduce the uncertainties in estimates from both models: studies at the lowest end of the global exposure distribution, but, most critically, studies from China, India, and other LMICs where levels remain very high. As these new studies are published, their results will be added to both the IER and GEMMs.

## 6. Conclusions

Over the past decade considerable advances have been made in methods for estimating mortality attributable to outdoor $PM_{2.5}$. Current relative risk estimators use the available epidemiologic evidence in different ways and under different assumptions and produce varying estimates of attributable mortality. Additional research is needed to reduce this uncertainty. Analysts need to carefully consider the available estimators with respect to their analytic objectives and specific circumstances. At this point in time and given likely future developments in the current risk models, we suggest that global estimates use the IER cause-specific estimator and that non-accidental cause of death GEMMs should only be applied after careful consideration of the distribution of causes of death in the target population of interest.

**Author Contributions:** R.B. and A.C. equally contributed to all aspects of this work. All authors have read and agreed to the published version of the manuscript.

**Funding:** A.C. was supported by the Health Effects Institute.

**Conflicts of Interest:** The authors declare no conflict of interest.

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
