# Peer review of "Relative Risk Functions for Estimating Excess Mortality Attributable to Outdoor PM2.5 Air Pollution: Evolution and State-of-the-Art"

_atmosphere, doi:10.3390/atmos11060589_

Round 1
Reviewer 1 Report
This manuscript discussed various practical issues related to the use of risk models to estimate disease burdens attributable to ambient PM2.5 pollution. Risk models are widely applied to support the development of environmental policies and/or interventions to reduce PM pollution. With an increasing number of cohort studies and the development of new fitting methods, the risk models have evolved rapidly. Thus, this work is important as a guide to ensure the proper use of these methods in future studies. This manuscript is well-written and in a good structure. I think this paper should be published in the Atmosphere.
Minor comments:
The actual and counterfactual PM2.5 exposures were defined as Zc and ZF in lines 109-110, but as PM2.5 and cf in line 176. Suggest unifying their uses. If the difference is due to a constraint in the values of cf, it should be clearly stated. Similar comments apply for line 208.
Since this study only focused on PM2.5, it would be more accurate to use the term “outdoor PM2.5 pollution” in various places instead of “outdoor air pollution”, unless the statement applies to ozone as well.
Author Response
Thanks very much for your comments.
Response: We have changed ZF to cf and defined cf as the counterfactual
Since this study only focused on PM2.5, it would be more accurate to use the term “outdoor PM2.5 pollution” in various places instead of “outdoor air pollution”, unless the statement applies to ozone as well.
Response: We have changed “ambient air pollution” to “outdoor fine particulate air pollution” throughout the manuscript.
Reviewer 2 Report
I appreciate the work that the authors did to describe the current practices, relative to the history, of modeling risk functions to estimate excess mortality due to PM2.5 globally. The authors do a good job presenting the LL, IER, and GEMM approaches along with their strengths and limitations.
The progression is clear and the model equations and descriptions are adequate with the proper citations.
My only suggestion would be to consider adding a figure showing where in the PM2.5 exposure distribution the coverage for each method changes/is appropriate. For example, showing how the LL model works well up until 30uq/m3, how GEMM can extend coverage to higher concentrations, and how the IER has the most flexible shape, while somewhat changing how PM exposure is measured.
Author Response
Thanks very much for your comments.
Response: We have added a new subsection 4.2.1 illustrating the relative risk predictions over both the lower concentration range observed in Western High Income countries and over the entire global concentration range. We did this for the LL and GEMM, for all non-accidental deaths, and the GBD2017 IER for six specific causes of death. We also included plots of the PAF for all three models over both low and global exposure ranges. To illustrate the age dependency modelled by both the GEMM and IER, we examined two age groups, 50-54 year olds and 80-84 year olds.